# Half-Fluence, Half-Dose Photodynamic Therapy: Less Direct Damage but More Inflammation?

**DOI:** 10.3390/ph16040494

**Published:** 2023-03-27

**Authors:** Thomas Desmettre, Martin A. Mainster, Gerardo Ledesma-Gil

**Affiliations:** 1Centre de Rétine Médicale, 187 rue de Menin, 59520 Marquette-Lez-Lille, France; 2Department of Ophthalmology, University of Kansas School of Medicine, Prairie Village, KS 66208, USA; 3Retina Department, Institute of Ophthalmology, Fundación Conde de Valenciana, Mexico City 06800, Mexico

**Keywords:** bacillary layer detachment, central serous chorioretinopathy, choroidal effusion, choroidal hyperpermeability, neovascular age related macular degeneration, optical coherence tomography, photodynamic therapy

## Abstract

**Objective:** To present clinical findings and multimodal imaging of three patients who developed bacillary layer detachments (BALADs) shortly after half-fluence, half-dose (HFHD) verteporfin photodynamic therapy (PDT). **Methods:** Retrospective observational case series. Three patients were treated with HFHD-PDT for (1) macular neovascularisation five years after resolved central serous chorioretinopathy (CSC), (2) persistent serous retinal detachment (SRD) from chronic CSC, and (3) neovascular age-related macular degeneration with persistent SRD despite intravitreal anti-VEGF therapy. **Results:** Each patient developed a BALAD after HFHD-PDT. Acute fulminant exudation caused subretinal fluid expansion into the inner photoreceptor layer, cleaving myoid from ellipsoid zones in the central macula. Subretinal fluid and the BALADs subsequently resolved over 6–8 weeks. **Conclusions:** The subretinal fluid and BALAD following HFHD-PDT were transient and did not cause photoreceptor damage over a 6-month follow-up period. We speculate that the reduced-impact HFHD protocol decreases direct tissue damage but increases proinflammatory cytokines. The long-term pathophysiological consequences of the resolved BALADs are unknown.

## 1. Introduction

Standard photodynamic therapy (PDT) involves intravenous injection of 6 mg/m^2^ verteporfin followed by 50 J/cm^2^ fluence infrared laser (690 nm) irradiation of the chorioretinal treatment site [1,2]. PDT was initially used for treating neovascular age-related macular degeneration (nAMD) and pathological myopia, conditions now managed primarily with anti-vascular endothelial growth factor (anti-VEGF) therapy [3,4,5]. Currently, PDT is mostly used for treating chronic central serous chorioretinopathy (cCSC), polypoidal choroidal vasculopathy (PCV), and choroidal hemangiomas [6]. It is also occasionally used in combination with anti-VEGF therapy to treat nAMD resistant to anti-VEGF monotherapy [7,8]. PDT was introduced for treating cCSC in the early 2000s and validated in subsequent clinical trials [9,10].

Reduced dose and fluence PDT protocols were developed in the mid-2000s in the hope of decreasing adverse PDT effects including choroidal ischemia with transient vision loss, retinal pigment epithelium (RPE) atrophy, and macular neovascularization (MNV) [11]. Initially, half-fluence or half-dose protocols were used independently and, later, they were combined into a half-fluence, half-dose (HFHD) protocol [12,13]. Recently, verteporfin production in the United States has been paused, creating a worldwide shortage of the drug [14]. This shortage issue has had a strong impact on therapeutic options across the globe with restricted use of PDT in some locations and increased use of HFHD procedures in other locations to preserve available verteporfin supplies.

The HFHD-PDT protocol aims to preserve the beneficial effects of PDT while minimizing associated chorioretinal damage. Unfortunately, a reduced photosensitizer-effect protocol can cause unexpected and undesired increases in post-operative exudation and even bacillary layer detachments (BALADs) have been described in patients treated for cCSC [15,16,17,18,19,20,21]. This problem may not be observed unless patients report their post-procedural vision problems because follow-up visits typically are not scheduled until 6–8 weeks after PDT. We have now observed HFHD-PDT-related increased exudation in patients treated for cCSC [18] and others treated for MNV associated with cCSC and for nAMD unresponsive to anti-VEGF monotherapy. 

## 2. Results

### 2.1. Case 1

A 60-year-old man presented with SRD and MNV 5 years after previously resolved cCSC. Six monthly intravitreal anti-VEGF (aflibercept) injections reduced exudation but the persistence of a SRD prompted a change in treatment and the choice of PDT. Six weeks after the last aflibercept injection, HFHD-PDT was performed with a laser spot size of 2000 µm. Preoperative visual acuity was 20/100. Fluorescein and OCT-angiography at that time revealed MNV at the superior-nasal border of the fovea and structural OCT imaging documented the associated SRD (Figure 1).

Two days after PDT, the patient returned complaining of decreased vision in his right eye. Visual acuity was 20/200. OCT showed increased subretinal fluid (SRF) extending to the central fovea where there was a central BALAD (Figure 2). The BALAD resolved in two weeks, with an improvement in the patient’s vision to 20/100. SRF took an additional 6 weeks to regress completely. The patient has been followed for 10 months without recurrence of exudation or other retinal problems.

### 2.2. Case 2

A 56-year-old man was referred for metamorphopsia and decreased vision in his right eye persisting for several months. His visual acuity was 20/100. Multimodal imaging showed active cCSC. His symptoms persisted for three more months and the patient was treated by performing HFHD-PDT with a spot size of 3000 µm. 

The patient returned two days after PDT complaining of markedly decreased vision in his right eye. Visual acuity had decreased there to 20/250. Structural OCT showed a foveal BALAD, a small central RPE detachment, and increased SRF extending from the optic disc to the temporal macula (Figure 3). The patient’s visual symptoms resolved over the next five days, along with the BALAD and SRF. Two months later, no OCT abnormalities persisted except for a small area of foveal ellipsoid and interdigitation zone disruption.

### 2.3. Case 3

A 74-year-old woman had persistent SRF despite multiple anti-VEGF (aflibercept) injections for MNV due to nAMD in her left eye. HFHD-PDT (spot size 3000 µm) was performed six weeks after the last aflibercept injection in an effort to stop further SRF accumulation and visual decline. The patient presented three days after the PDT for an additional intravitreal aflibercept injection. She reported significant vision loss in her left eye where structural OCT now showed a prominent increase in SRF and a central BALAD (Figure 4). The BALAD resolved in 10 days and most SRF reabsorbed in the following 5 weeks.

## 3. Materials and Methods

Here, we report clinical and multimodal imaging findings of three patients treated with HFHD-PDT for (1) MNV from resolved cCSC, (2) persistent serous retinal detachment (SRD) from cCSC, and (3) nAMD with persistent SRD despite intravitreal anti-VEGF therapy.

The patient records and images were reviewed retrospectively and data abstracted for this report. All patients underwent multimodal imaging, including color, near-infrared and autofluorescence imaging, fluorescein angiography (Topcon TRC 50 DX, Topcon, Tokyo, Japan and Nidek Mirante, Nidek Co. Ltd., Gamagori, Japan), and optical coherence tomography (OCT) (Cirrus 5000 HD-OCT, Carl Zeiss Meditec, Dublin, CA, USA). The findings in Case 2 were discussed in an earlier report [18] but we have identified two additional cases since its publication. 

All visual acuities listed thereafter are best-corrected visual acuities expressed as Snellen ratios and measured at 5 m.

Each patient received half-fluence PDT (25 J/cm^2^) over 83 s using a 693 nm wavelength) Vitra PDT, Quantel Medical, Cournon-d’Auvergne, France) adapted on a slitlamp biomicroscope (Haag-Streit, Bern, Switzerland) 15 min after the infusion of a half-dose of verteporfin (Visudyne^®^, Cheplapharm Arzneimittel GmbH, Greifswald, Germany) (3 mg/m^2^) over 10 min. The PDT laser spot size varied according to the patient’s condition.

## 4. Discussion

Small, transient increases in exudation can occur in the first few days after PDT regardless of the retinal condition being treated [19,22,23], but prominent increases in exudation and BALAD formation after PDT are less common [15,16,24]. BALADs can occur in numerous inflammatory, infectious, neoplastic, degenerative, and toxic conditions [25,26,27]. Their origin has been attributed to hyperacute choroidal fibrinous exudation too fulminant for subretinal space containment cleaving the myoid from the ellipsoid zones of photoreceptor inner segments [18,22] when intraretinal mechanical forces exceed the tensile strength of the myoid zone [24].

PDT’s hyperproduction of reactive oxygen species (ROS) causes endothelial cell and collateral tissue damage [1,2]. Cellular, vascular, and immunological factors are involved [1,28,29,30,31]. Their relative effects depend on the target tissue, photosensitizer, and treatment parameters [31]. PDT causes plasma membrane and mitochondria damage, with the production of cytokines including VEGF and PDGF [2,29]. It can also damage RPE cells and their intercellular tight junctions, altering the outer blood retinal barrier at least transiently [32,33]. Transient alteration of the outer blood retina barrier could account partly for our patients’ increased post-operative SRF despite usage of the HFHD protocol designed to reduce PDT side effects. We speculate that reduced-impact HFHD protocols decrease direct tissue damage but increase pro-inflammatory cytokines causing our patients’ SRF escalation and the BALAD it caused

Exudation after PDT has been previosuly referred to as “transient serous retinal detachment” [23], “acute exudative maculopathy” [20], and “PDT-induced acute exudative maculopathy (PAEM)” [21]. Recent large case series studies of different designs have found different incidences of acute exudation following PDT [19,20,21]. In a retrospective case series of 155 eyes treated with standard-dose and standard- or half-fluence PDT, Manayath, et al. found that exudation occurred after 4.5% of all PDT sessions or 1.2%, 6.7%, and 8.5% of treatments for cCSC, type 1 MNV, and PCV, respectively [21]. In a retrospective case series of 58 eyes, Mammo and Forooghian found that standard-dose and half- or quarter-fluence PDT for cCSC or nAMD produced acute exudation after 1.4% of treatments [20]. Higher incidences of post-PDT exudation were reported by Fernández-Vigo et al. in a prospective study of 75 patients treated with standard-dose, half-fluence, 4000 µm spot size PDT for cCSC [19]. In that study, post-PDT acute exudation and BALADs were reported in 30% and 14% of treated eyes, respectively [19]. Neither retrospective case series report addressed post-operative BALAD. 

Manayath et al. identified PDT-induced acute exudative maculopathy only in patients treated with full-dose PDT and spots sizes ≥3500 µm. Fernández-Vigo et al.’s protocol limited them to 4000 µm spot sizes in their full-dose PDT cCSC treatment protocol. Unlike these studies, we used a half-dose, half-fluence protocol in an effort to minimize collateral PDT damage. We found that post-PDT exudative BALADs can occur with smaller 2000–3000 µm treatment spot sizes in half-dose PDT. Additionally, the Fernández-Vigo et al. study found BALADs after PDT therapy for cCSC patients, but we also identified them after PDT for MNV in nAMD and MNV that occurred without active CSC in an eye previously treated for cCSC (pachychoroid neovasculopathy [34]). 

In our three patients, increased exudation and BALAD formation was transient and did not cause photoreceptor damage in the post-operative monitoring period. The long-term effects of resolved BALADs, if any, have yet to be determined. Limitations of our study include its small size and retrospective design. We believe that patients undergoing HFHD-PDT should be informed that an infrequent but prominent decrease in vision can happen post-operatively and recommend that they return for further evaluation should that occur.

## Figures and Tables

**Figure 1 pharmaceuticals-16-00494-f001:**
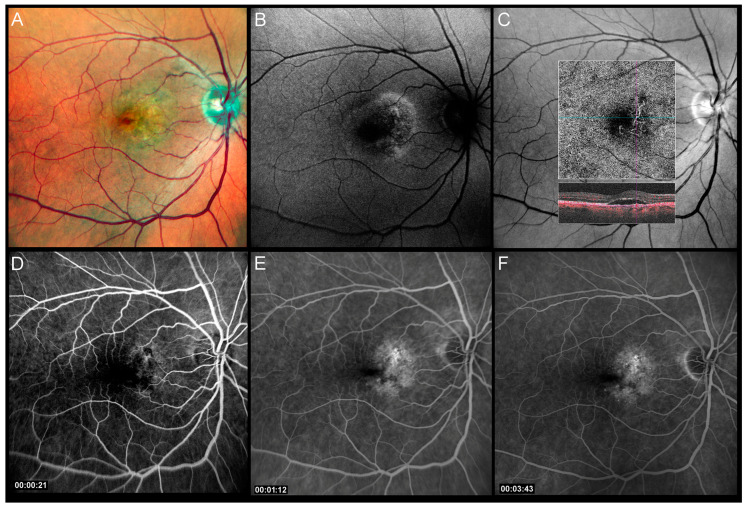
Preoperative multimodal imaging of patient 1: (**A**) Pseudocolor image shows pigmentary changes and thickening in the nasal part of the macula; (**B**) autofluorescence image, note the lobulated area with a pattern of speckled autofluorescence centrally and hyperautofluorescent borders; (**C**) green channel image with superimposed en face optical coherence tomography angiography and OCT B-scan showing the choroidal neovascularization; (**D**–**F**) progressive phases of the fluorescein angiogram which show slow dye diffusion confirming the presence of active macular neovascularization.

**Figure 2 pharmaceuticals-16-00494-f002:**
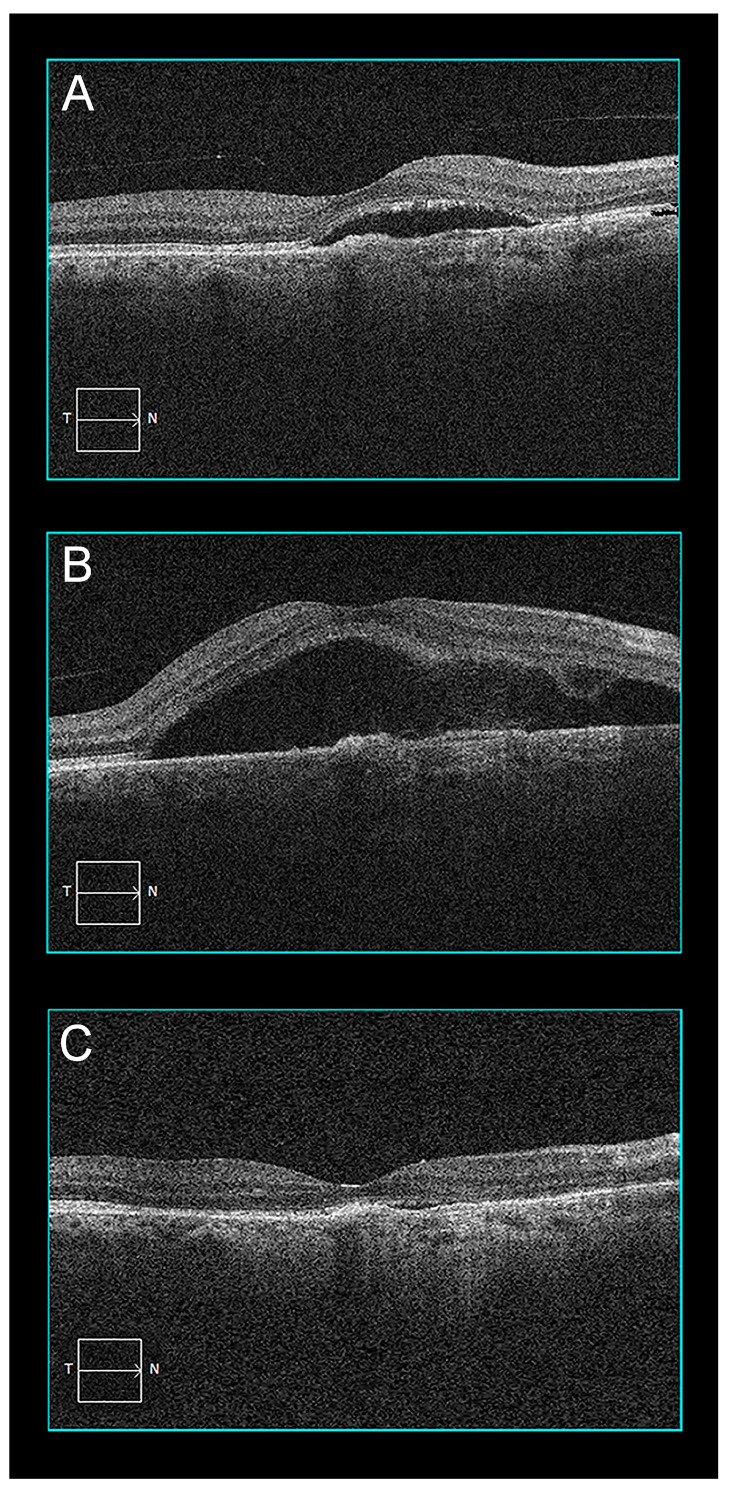
Patient 1 optical coherence tomography evolution before and after PDT: (**A**) Baseline OCT B-scan showing subretinal fluid and a shallow retinal pigment epithelium detachment; (**B**) two days after PDT, there was increased exudation with a bacillary layer detachment; (**C**) after 2 months, exudation resorption was almost complete.

**Figure 3 pharmaceuticals-16-00494-f003:**
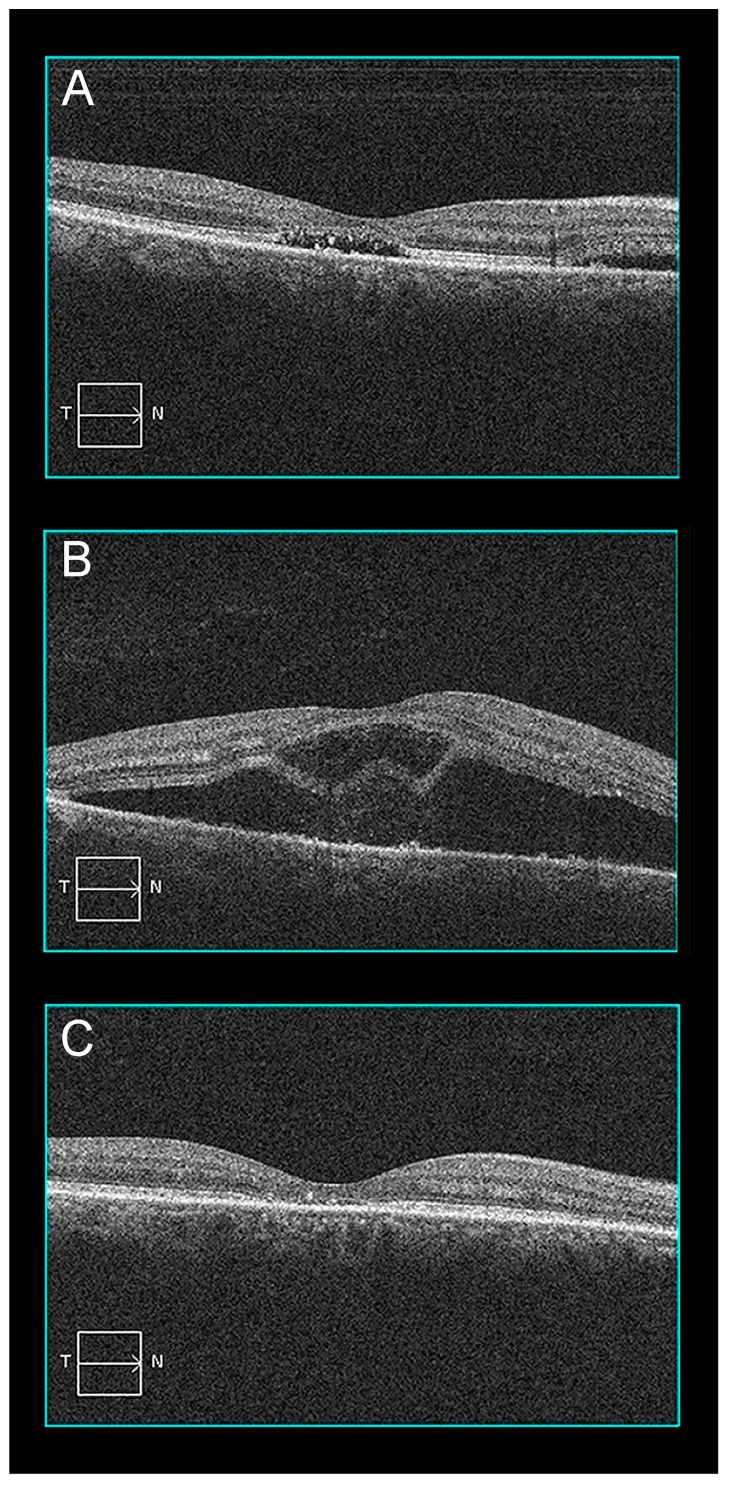
Patient 2 optical coherence tomography evolution before and after PDT: (**A**) Baseline OCT B-scan showing persistent subretinal fluid under the fovea and nasally; (**B**) two days after PDT, there was increased exudation with a bacillary layer detachment; (**C**) after 2 months, the exudation had reabsorbed completely but a small area of foveal thinning with ellipsoid and interdigitation zone disruption remained.

**Figure 4 pharmaceuticals-16-00494-f004:**
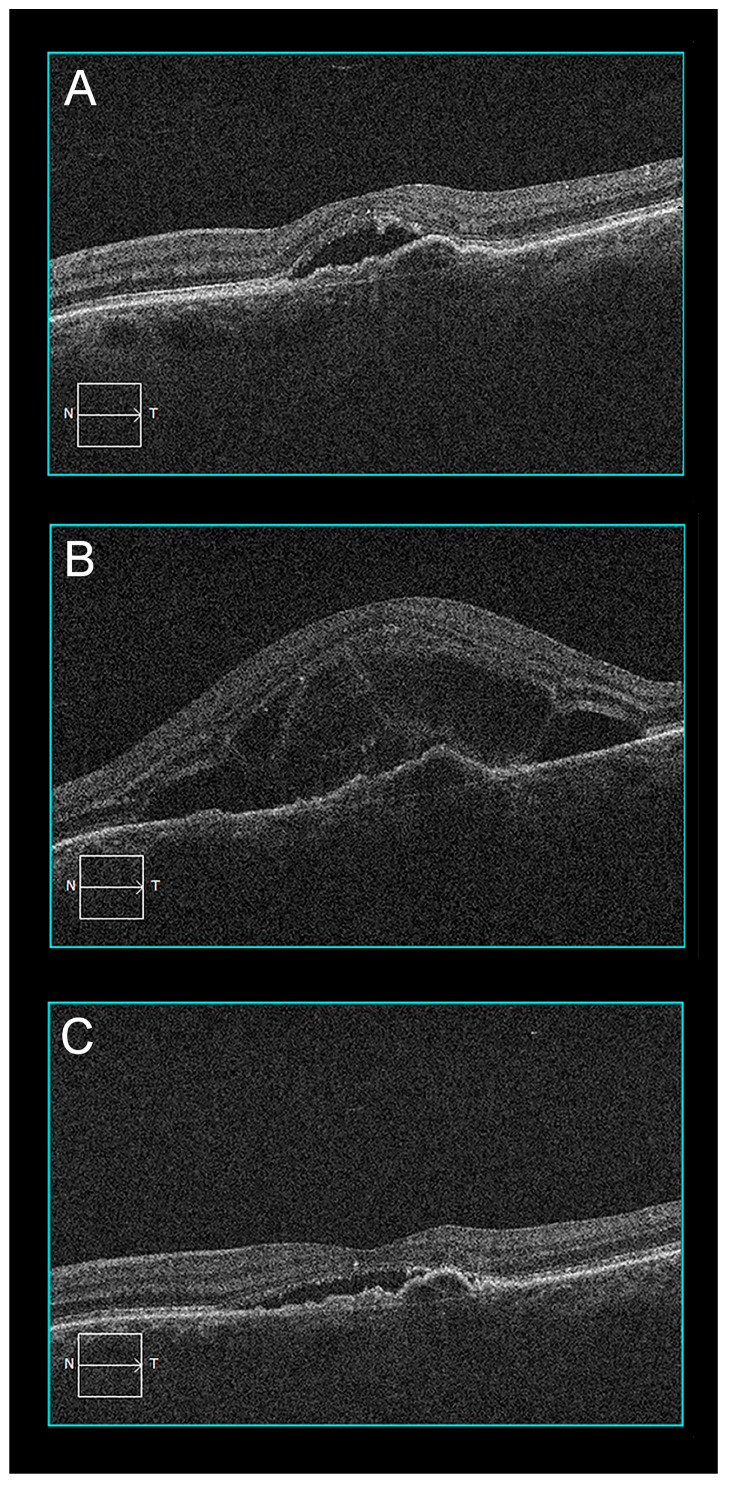
Patient 2 optical coherence tomography evolution before and after PDT: (**A**) Baseline OCT B-scan shows type 1 macular neovascularization with subretinal fluid; (**B**) two days after PDT, subretinal fluid increased substantially and a bacillary layer detachment was present; (**C**) after 2 months, there was a significant reduction of the subretinal fluid and the bacillary layer detachment had resolved.

## Data Availability

Data is contained within the article.

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
