# Peer review of "Half-Fluence, Half-Dose Photodynamic Therapy: Less Direct Damage but More Inflammation?"

_pharmaceuticals, 2023, doi:10.3390/ph16040494_

Round 1

Reviewer 1 Report

In this study, the authors presented cases of increased SRF with BALAD after receiving half-fluence, half-dose photodynamic therapy for MNV or CSC. The authors then speculated that reduced-impact HFHD protocols decrease direct tissue damage but increase proinflammatory cytokines. In fact, increased SRF after full dose and full fluence PDT is not infrequently seen. Furthermore, even no incidence of increased SRF and BALAD after half-fluence, half-dose PDT was presented in this manuscript. No evidence presented in this manuscript could show any hint for increased inflammation after half-fluence, half-dose PDT. 

Author Response

All our patients had massive exudation with serous retinal fluids (SFR) and a bacillary layer detachment (BALAD displayed on Optical Coherence Tomography (OCT) images. BALAD has been reported in numerous primary and secondary inflammatory chorioretinal conditions and the general consensus is that it is caused by inflammation.  Its presence in our patients represents  transient post-PDT inflammation. This notion of inflammation his coherent with the literature about “Photodynamic-therapy Acute Exudative Maculopathy (PAEM)” (1-3).

  1. Chan WM, Lam DS, Lai TY, et al. Choroidal vascular remodelling in central serous chorioretinopathy after indocyanine green guided photodynamic therapy with verteporfin: a novel treatment at the primary disease level. Br J Ophthalmol 2003; 87:1453–1458
  2. Iovino C, Au A, Chhablani J, et al. Choroidal anatomic alterations after photodynamic therapy for chronic central serous chorioretinopathy: a multicenter study. Am J Ophthalmol 2020;217:104–113.
  3. Fernández-Vigo JI,et al. Acute Exudative Maculopathy and Bacillary Layer Detachment in patients with Central Serous Chorioretinopathy after Photodynamic Therapy. Retina. 2022 May 1;42(5):859-866

Reviewer 2 Report

This is an interesting case series concerning the half-fluence, half-dose photodynamic therapy utilized for 3 patients with 3 different retinal diseases.

This article is well-organized and deals with a really challenging topic.

However, some aspects of this article should be improved:

1) a minor revision of the English throughout the manuscript should be performed to improve the readability of the article.

2) In the Methods section, the authors should deeply describe how they performed PDT.

Author Response

We thank the reviewer for these nice comments. 

Our manuscript has been entirely reviewed by a native English speaker and we defer to reviewer 4’s unambiguous observation that our manuscript “is an extremely well-written case series with elegant and adequate discussion of clinical and pathophysiological relevance”.

Regarding the Methods section, the PDT procedure had been described quite precisely (fluence, dose of visudyne, delay for laser irradiation, spot size..). However, we added a comment for non-ophthalmologist readers that retinal PDT is performed with the laser adapted on a slitlamp biomicroscope (Haag-Streit, Bern, Switzerland). 

Reviewer 3 Report

it is a interesting cases! We don t know what will happend after long time.

I don t understant wen they administrt aflibercept before or after or paralel

Author Response

We thank the reviewer for these remarks.

Aflibercept was used for patients #1 and #3 prior to PDT.

For patient #1, as mentioned in the manuscript, 6 monthly aflibercept intravitreal injections had been administered to treat CSC-related choroidal neovascularization with a serous retinal detachment (SRD).  When this treatment failed to completely resorb the SRD, we performed PDT (6 weeks after the last aflibercept injection). The patient’s resultant BALAD decreased over 2 weeks with remaining exudation eventually resolving.

We modified our description of case #1 to clarify why aflibercept therapy had been performed. At the end of case #1 description we commented that currently the patient has been followed for 10 months.

For patient #2, in the same way we added that PDT was performed six weeks after the last aflibercept injection in an effort to stop further serous retinal fluids (SRF) accumulation and visual decline.

Reviewer 4 Report

In this paper, Desmettre et al. presented three cases of bacillary layer detachment shortly after half-fluence half-dose PDT treatment. I congratulate the authors for their work.

I have to underscore that this is an extremely well-written case series with elegant and adequate discussion of clinical and pathophysiological relevance. I only have few minor comments:

1. Introduction: Regarding the verteporfin shortage, please briefly mention that the ongoing shortage issue has an strong impact on the availability of therapeutic options across the globe (PMID: 35388619). Using reduced protocols, such as the HFHD as presented here, is the only option available in many centers including my own.

2. Methods: Regarding this phrase "[Visudyne® ; Novartis AG, MI]". The name of the city and the country should be stated.

3. Methods: For clarity, please state that all visual acuities were the best-corrected visual acuity, the distance at which they were measured, and that they are presented in Snellen ratios.

4. Discussion: Regarding the very last sentence "We believe that patients undergoing HFHD-PDT should be informed that a prominent decrease in vision can occur post- operatively and counseled to return for further evaluation should it occur.". I suggest that this phrase is made more precise to emphasize that this is a rare phenomenon and only occurs on some patients.

Author Response

We would like to thank the reviewer for these helpful comments.

  1. We have modified the introduction section to briefly emphasize difficulties in treating patients due to verteporfin shortage and have added a reference to Sirks M. et al (2022) to better illustrate the impact of the shortage on ophthalmologists' practice.
  2. "[Visudyne® ; Novartis AG, MI]" has been replaced by “[Visudyne®, Cheplapharm Arzneimittel GmbH, Greifswald, Germany]
  3. We documented in the Methods section that all visual acuities are best-corrected visual acuities expressed as Snellen ratios and measured at 5m.
  4. We modified the last sentence to read: “We believe that patients undergoing HFHD-PDT should be informed that an infrequent but prominent decrease in vision can happen post-operatively and recommend that they return for further evaluation should that occur.”

Round 2

Reviewer 1 Report

As the authors answered, increased SRF and BALAD are common after PDT. Although this is an interesting case, it is not reasonable to deduct that reduced-impact HFHD protocols decrease direct tissue damage but increase pro-inflammatory cytokines from 3 cases.

Author Response

Reviewer 1 states erroneously that we mentioned: "increased SRF and BALAD are common after PDT".  That statement is incorrect.  In our paper we describe 3 cases with different conditions, all producing the same uncommon massive exudation after HFHD-PDT. 

Reviewer 1 also states: “Although this is an interesting case, it is not reasonable to deduct that reduced-impact HFHD protocols decrease direct tissue damage but increase pro-inflammatory cytokines from 3 cases.” Assuming Reviewer 1 was trying to use the word “deduce” (rather than the incorrect word “deduct”), he/she is again erroneous.  We never claimed to “deduce” that reduced impact HFHD protocols were the cause of the inflammation that occurred. We stated clearly that we “speculate” that it was that cause. Indeed, that is plainly acknowledged even in the title of our manuscript which ends with a “?” and in the last paragraph of our manuscript which states plainly: “Limitations of our study include its small size and retrospective design”.      

Reviewer 1’s first comment of our manuscript was based on his/her failure to understand that the exudation we documented was caused by inflammation.  We explained that fact to him/her in our response. Now Reviewer 1 argues that we do not have enough cases to “deduct” (should be “deduce”) an etiology for the inflammation.  We state plainly that we agree, previously acknowledging that the small number of cases is a study limitation and carefully using the word “speculate” rather than “deduce”.

Reviewer 3 Report

Interesting cases. good results!

Author Response

Thank you very much!